# Development of TRIB3-Based Therapy as a Gene-Independent Approach to Treat Retinal Degenerative Disorders

**DOI:** 10.3390/ijms25094716

**Published:** 2024-04-26

**Authors:** Trong Thuan Ung, Christopher R. Starr, Assylbek Zhylkibayev, Irina Saltykova, Marina Gorbatyuk

**Affiliations:** Department of Optometry and Vision Science, University of Alabama at Birmingham, Birmingham, AL 35233, USA; ttung@uab.edu (T.T.U.); crstarr@uab.edu (C.R.S.); askokshe@uab.edu (A.Z.);

**Keywords:** retinal degeneration, unfolded protein response, TRIB3, cell-penetrating peptides, afatinib

## Abstract

Inherited retinal degeneration (RD) constitutes a heterogeneous group of genetic retinal degenerative disorders. The molecular mechanisms underlying RD encompass a diverse spectrum of cellular signaling, with the unfolded protein response (UPR) identified as a common signaling pathway chronically activated in degenerating retinas. TRIB3 has been recognized as a key mediator of the PERK UPR arm, influencing various metabolic pathways, such as insulin signaling, lipid metabolism, and glucose homeostasis, by acting as an AKT pseudokinase that prevents the activation of the AKT → mTOR axis. This study aimed to develop a gene-independent approach targeting the UPR TRIB3 mediator previously tested by our group using a genetic approach in mice with RD. The goal was to validate a therapeutic approach targeting TRIB3 interactomes through the pharmacological targeting of EGFR-TRIB3 and delivering cell-penetrating peptides targeting TRIB3 → AKT. The study employed rd10 and P23H RHO mice, with afatinib treatment conducted in p15 rd10 mice through daily intraperitoneal injections. P15 P23H RHO mice received intraocular injections of cell-penetrating peptides twice at a 2-week interval. Our study revealed that both strategies successfully targeted TRIB3 interactomes, leading to an improvement in scotopic A- and B-wave ERG recordings. Additionally, the afatinib-treated mice manifested enhanced photopic ERG amplitudes accompanied by a delay in photoreceptor cell loss. The treated rd10 retinas also showed increased PDE6β and RHO staining, along with an elevation in total PDE activity in the retinas. Consequently, our study demonstrated the feasibility of a gene-independent strategy to target common signaling in degenerating retinas by employing a TRIB3-based therapeutic approach that delays retinal function and photoreceptor cell loss in two RD models.

## 1. Introduction

Inherited retinal degeneration (RD) is a heterogeneous group of genetic disorders that affect the retina. This condition is typically characterized by the progressive deterioration of the retina’s neuronal cells, particularly the photoreceptors. However, the conditions can vary in their inheritance patterns, affected genes, age of onset, rate of progression, and specific symptoms. While the molecular mechanisms of degenerating retinas may include a diverse spectrum of cellular signaling, the unfolded protein response (UPR) has been recognized as a common signaling pathway chronically activated in various animal models of RD.

Tribbles homolog 3 protein (TRIB3) has been identified as a key mediator of the PERK UPR arm. During ER stress, its induction leads to ATF4 inhibition and, therefore, the regulation of the ATF4-dependent downstream gene transcription network. The multifaceted role of TRIB3 has been previously highlighted in the literature [1]. Thus, TRIB3 has been proposed to serve as a metabolic stress indicator involved in the regulation of a myriad of varied metabolic pathways, including insulin signaling, lipid metabolism, and glucose homeostasis. In particular, TRIB3 acts as an AKT pseudokinase, prevents the phosphorylation of AKT upon binding, and abrogates AKT interactions with its partners [2].

The AKT → mTOR axis is involved in a wide range of cellular processes, such as glucose metabolism, cell proliferation, survival, and protein synthesis [3]. Both AKT and mTOR play pivotal roles in the mechanism of retinal pathogenesis. For instance, the inactivation of the AKT survival pathway during photoreceptor degeneration in mice has been reported [4,5,6], and the enhancement of AKT has been proposed as a neuroprotective strategy for dying retinal cells [2,7,8]. mTOR activity has been considered essential in healthy and diseased retinas. It has been reported to affect RPE cell survival [9] and cone function [10], associated with metabolic dysfunction and contributing to their cell death. Furthermore, a decline in phosphorylated mTOR coincided with cone degeneration [11]. Moreover, strategies such as the constitutive activation of mTORC1 via the disruption of TSC1 significantly improved retinal cell metabolism and survival [12]. Additionally, targeting the downstream mTORC1 effector S6K1 by subretinal adeno-associated virus (AAV)-mediated S6K1 delivery significantly improved rod photoreceptor survival in rd10 mice, a model of retinal degeneration expressing mutant PDE6β [13].

Our recent work with rd16 mice mimicking inherited RD demonstrated that TRIB3 is upregulated in their retinas and its ablation modifies the activity of the AKT → mTOR axis [5,14]. Furthermore, TRIB3 deficiency in the retina of rd16 mice leads to photoreceptor cell survival and an improvement of retinal function [14]. The research conducted by our group further investigated the neuroprotection mechanism of a TRIB3-based strategy in rd16 mice. We observed that the activation of the AKT → mTOR axis led to a significant increase in the translational rate in degenerating retinas. This increase resulted in the synthesis of viable proteins, possibly contributing to the preservation of retinal function [14]. Thus, in that study, we observed that the boost in autophagosome formation occurred concomitantly with an increase in rhodopsin protein levels and the elevation of E3 ligase Parkin1. Based on the results, we then proposed that modulation of TRIB3 levels may retard retinal degeneration and that TRIB3 may be a promising therapeutic target to treat various retinal degenerative disorders.

The development of new strategies for the treatment of inherited RD is a focus of current biomedical research. Among strategies targeting mutant photoreceptor-specific genes are adeno-associated and lentiviral-based therapies with the options of either gene augmentation or silencing [15] and regenerative therapy using photoreceptor cells or retinal sheet transplantation [16]. While gene therapy for patients expressing aberrant photoreceptor-specific proteins holds great promise [17], the development of a gene-independent approach targeting common cellular signaling could be an attractive alternative option. In this regard, TRIB3, the UPR mediator previously tested by our group using a genetic approach in mice, could be an excellent candidate to develop a multifaceted gene-silencing method for further validation in different animal models of RD.

In this study, we validated a therapeutic approach targeting TRIB3 and TRIB3 interactomes in the mouse retina. For example, previous studies have demonstrated that TRIB3 interacts with EGFR, enhancing EGFR recycling, stability, and downstream activity. Additionally, TRIB3 has been shown to reduce the activity of AKT through binding to this protein [2,12]. Using two models of RD, rd10 and P23H RHO, we employed a pharmacological strategy targeting EGFR-TRIB3 and the delivery of cell-penetrating peptides targeting TRIB3 → AKT interactomes. The application of both strategies reduced TRIB3 activity and significantly mitigated retinal degeneration in these mice.

## 2. Results

### 2.1. Targeting TRIB3-AKT in Mice Alleviates Retinal Degeneration

The previous study with rd16 mice showed that TRIB3 ablation delays retinal degeneration [14]. To expand on this, we tested whether this approach could postpone the onset of retinal dystrophy in another model of RD, the P23H RHO mice. Our scotopic ERG analysis at p45 revealed preserved a- and b-waves in P23H RHO TRIB3^−/−^ mice compared to the decline observed in controls. These results provide the rationale for the further exploration of therapeutic options targeting TRIB3 in P23H RHO retinas (Figure 1A). Cell-penetrating peptides (CPPs) targeting TRIB3-AKT interactomes have recently been developed [2]. It has been demonstrated that Pep2–Ae binding to TRIB3 enhanced the activity of AKT1 in MEC- and PDX-derived breast tumors [2]. Therefore, we decided to test whether the Intravitreal (IVT) injection of rd10 mice with Pep2-Ae leads to an increase in p-AKT levels in mouse retinas (Figure 1B). The treatment of mice with Pep2-Ae demonstrated that normalized p-AKT was increased 1.5-fold in the retina 6 h postinjection as compared to control Pep2 CPP. Therefore, next, we decided to validate CPPs in mice with RD. 

We previously found that the p-AKT level is reduced in the retinas of P23H RHO rats [18]. Moreover, the overexpression of AKT3 in rd10 mice has been reported to be neuroprotective [8]. Therefore, we next injected the P23H RHO mice with CPPs and followed the decline in their retinal function. At p60, we observed the preservation of the scotopic a- and b-wave amplitude in the eye injected with Pep2-Ae as compared to the control injected eye (Figure 1C). An analysis of cone-dominant ERG responses did not reveal a statistically significant difference between the experimental and control injected retinas at p60.

Afatinib is a FDA-approved EGFR inhibitor, the treatment of which promotes the degradation of TRIB2, another TRIB family protein and an EGFR binding partner [19]. On the other hand, similar to TRIB2, TRIB3 binds to EGFR and stabilizes its recycling, facilitating the development of cancer [20]. These data suggest that the extensive activity of TRIB3 in the retina could increase the stability of EGFR. Knowing that excessive TRIB3 downregulates p-AKT and its ablation is neuroprotective for mice with RD, we, next, wondered whether the treatment of degenerating retinas with afatinib leads to the reduction of TRIB3 in P23H RHO retinas.

Retinal tissues from p45 P23H RHO mice were isolated and cultured in medium supplemented with 100 μM afatinib for 48 h (Figure 1D). This treatment led to an approximately 50% reduction in retinal TRIB3 protein levels under these experimental conditions. These findings suggest that therapies aimed at targeting the TRIB3 protein and TRIB3-AKT interactome may offer benefits to degenerating retinas.

### 2.2. Treatment of 661W Cells with Afatinib Reduces EGFR and TRIB3

Given that the treatment of P23H RHO retinas with afatinib resulted in a decrease in TRIB3, we proceeded to treat cone-derived 661W cells with 5 μM afatinib in the next step. The results of the treatment indicated a reduction not only in the EGFR level, but also a diminishment of TRIB3 in the treated cells (Figure 2A). A two-fold reduction of TRIB3 was detected 24 h post treatment of 661W cells with afatinib (*p* < 0.05). To confirm these results, we transfected 661W cells with pTRIB3-GFP and treated cells with afatinib to calculate the fluorescence signal emitted from cells expressing fused TRIB3 by confocal microscopy. A two-fold decline in the fluorescence was observed in cells treated with afatinib as compared to the control (Figure 2C,D, *p* < 0.01). Taken together, these findings suggest that afatinib reduces TRIB3 levels in vitro, thus corroborating the results observed with the treatment of P23H RHO retinas.

### 2.3. Treatment of rd10 Mice with Afatinib Reduced TRIB3 and Prevented Retinal Function Loss

We then administered afatinib in rd10 mice from p15 to p25 by intraperitoneal (IP) injection. Thus, we learned that both EGFR and TRIB3 are reduced in the retinas of treated mice from 50% to 40% (Figure 2B, *p* < 0.05). These data served as a rationale to inspect the retinal degeneration of mice treated with afatinib.

The daily treatment of rd10 mice was conducted from p15 until the end of the experiment (p45). We found that both scotopic (Figure 3A–C) and photopic (Figure 3D–F) ERG amplitudes were elevated in rd10 mice treated with afatinib compared to vehicle-treated mice at p30. At p45, the effect was sustained, and, while untreated mice continued manifesting retinal degeneration, the treated group still showed an increase in scotopic (Figure 4A–C) and photopic (Figure 4D–F) ERG amplitudes. These data suggest that a delay in retinal function decline might be associated with the preservation of photoreceptor cell death in treated mice. Therefore, we, next, performed the histological analysis to assess the photoreceptor cell numbers in degenerating retinas.

### 2.4. Treatment of rd10 Mice Prevents Photoreceptor Cell Loss and Increases PDE6β

At p30, eyes were harvested from treated and control mice and cryosectioned. In H&E-stained sections, we found that rows of photoreceptor cells were significantly preserved across the entire retina in treated vs. untreated mice (Figure 5A,B). Moreover, by labelling with fluorescently coupled peanut agglutinin, we determined that the number of cone photoreceptor cells was significantly increased in the treated retinas (Figure 5C–E). Further, counting cones labeled with peanut agglutinin (PNA) in a flat-mounted retina, we found about a 30% increase in treated as compared to untreated mice.

We then performed the detection of PDE6β in treated retinas with IHC and found that their retinas manifest enhanced staining (Figure 6A and Appendix A). These data were in agreement with the results on total PDE activity in treated retinal extracts. Using GTP as a substrate, we identified that treated retinas manifest about a 30% higher PDE activity than untreated retinas (Figure 6B). Overall, these data demonstrate that treated retinas manifest an improved phototransduction signaling associated with photoreceptor cell survival, improved electrophysiology, and enhanced PDE6β activity. In concert with these results, we observed robust staining with the anti-RHO antibody in the retinas of mice treated with afatinib (Figure 6C).

## 3. Discussion

In this study, we validated a TRIB3-based gene-independent approach to treating retinal degenerative disorders associated with chronic UPR activation. Using two mouse models of RD with different etiologies and rates of retinal degeneration, along with two therapeutic approaches (targeting TRIB3 levels via pharmacological intervention and TRIB3 pseudokinase activity via CPP delivery to the retina), we delayed the onset of retinal degeneration. These strategies prevented photoreceptor cell death and functional decline in mice.

Rd10 mice manifest progressive autosomal recessive retinitis pigmentosa. They express a missense mutation in the catalytic β subunit of the PDE6 holoenzyme, and, when raised under normal light conditions with a 12 h light/12 h dark cycle, these mice show the complete degeneration of the photoreceptor outer nuclear layer (ONL) by p45 [21]. Furthermore, at p15, they manifest a reduction in all three PDE6 subunits (α, β, and γ) [21], leading to rapid rod photoreceptor degeneration, followed by cone degeneration [22]. Previous attempts to rescue these mice included gene therapy [22], housing under dark-reared conditions [21], treatment with epigenetic drug [23], cGMP analogues CN238 [24] and Rp-8-Br-PET-cGMPS [25], and trans-scleral electrical stimulation [26]. The P23H RHO mice degenerate at a relatively slower pace than the rd10 mice and have been the focus of therapeutic interventions as well. Approaches such as mutant RHO cut-and-replacement or gene editing have primarily been tested in these mice [27].

Both retinal functional loss and photoreceptor cell death were postponed in the mice with RD. The reduction in TRIB3 led to improvements in scotopic and photopic ERG amplitudes. The P23H RHO TRIB3^−/−^ mice exhibit an increase in both a- and b-wave amplitudes. Additionally, the increase in ERG amplitudes in the rd10 mice correlated with a decreased rate of photoreceptor cell death. Our previous report indicated that TRIB3 ablation restored p-AKT → p-mTOR activity, leading to an increase in vital protein synthesis [14]. In the current study, we provided evidence that pharmacologically targeting TRIB3 results in reduced TRIB3 protein levels in P23H RHO retinas and delays retinal degeneration in rd10 mice. Consequently, decreasing TRIB3 was associated with improved PDE activity and a more robust PDE6β state in the rd10 retinas. Rd10 photoreceptors appeared healthier in the afatinib-treated mice. As proof, we found an increase in rhodopsin in the treated rd10 retina. However, the latter raises the question of whether this increase benefits rd10 degenerating photoreceptors.

A recent study conducted with rd10 mice demonstrated that light activates a non-canonical pathway in rd10 retinas mediated by rhodopsin, contributing to retinal degeneration [21]. This activation is independent of transducing and PDE6 and causes photoreceptor cell death. Bearing this in mind, it is possible that increased RHO limits the therapeutic effect of afatinib in rd10 mice. In our study, we did not validate a combination of afatinib treatment and dark-rearing conditions. Therefore, this presents a limitation of the study that needs to be overcome in the future.

TRIB3 binds to EGFR and maintains its recycling. Previously, it has been shown that afatinib promotes the rapid degradation of TRIB2, another binding partner of EGFR [19]. Our data with 661W cells transfected with TRIB3-GFP and treated with afatinib (Figure 2) provide strong evidence that afatinib affects the level of TRIB3 too. Using confocal microscopy and western blotting, we demonstrated a reduction in fused TRIB3. While we cannot definitively conclude whether afatinib directly binds to TRIB3 or reduces its level through the degradation of the EGFR-TRIB3 complex at this point, these data, along with the findings from P23H RHO retinal explants treated with afatinib, suggest the potential use of afatinib for pharmacologically targeting TRIB3 in degenerating retinas. However, future experiments should focus on elucidating the mechanism behind an afatinib-induced TRIB3 reduction, particularly whether it involves enhanced TRIB3 degradation. Another limitation of our study is that we did not validate a combination of the two approaches (intraocular delivery of CPPs and treatment with afatinib). Given that the P23H RHO mice demonstrate a decline in retinal function associated with an increase in p-AKT in vivo and that a boost of AKT activity benefits rd10 mice [8], it would be interesting to see whether a combination of the two approaches strengthens the therapeutic effect, providing long-lasting benefits.

In summary, our study demonstrated the feasibility of a gene-independent strategy to target common signaling in degenerating retinas, employing a TRIB3-based therapeutic approach that resulted in the delay of retinal function and photoreceptor cell loss in two RD models. Further testing of this strategy in additional RD animal models would indeed contribute to a more comprehensive evaluation of this TRIB3-based therapeutic effect.

## 4. Materials and Methods

### 4.1. Animals and Treatments

Male and female rd10, P23H RHO, TRIB3 KO, and C57BL/6J mice were used in the study and were housed at the UAB (University of Alabama at Birmingham, AL, USA) animal core facility with a 12 h light/dark cycle and free access to a standard diet and water. Mice were euthanized by CO_2_ asphyxiation followed by cervical dislocation according to a protocol (IACUC#22378) approved by the UAB IACUC (Institutional Animal Care and Use Committee) committee.

Afatinib (CAS NO: 439081-18-2, LC Laboratories Woburn, MA, USA) was dissolved in DMSO and administrated to rd10 mice daily by intraperitoneal injection from p15 to p25 or from p15 to p45 in a dose of 15 mg/kg of body weight. PBS injections served as a control. Cell-penetrating peptides Pep2-con (Pep2-AAHTAAENRVAAA) and Pep2-Ae (Pep2-VAHTLTENRVLQN) were previously designed [20] and synthesized by Lifetein (Somerset, NJ, USA). The intravitreal (IVT) injection of CPPs at a dose of 5 μg was conducted once in rd10 (p5) and twice (p15 and p25) in P23H RHO mice.

### 4.2. Electroretinography (ERG)

To evaluate rod and cone function, ERG was performed using the UTAS BigShot (LKC technologies, Gaithersburg, MD, USA) instrument as previously described [14]. Briefly, mice were dark-adapted overnight and anesthetized with an intraperitoneal injection of ketamine/xylazine cocktail based on their body weight (100 mg/kg ketamine and 10 mg/kg xylazine). After pupil dilation with topical 2.5% phenylephrine (Paragon Bioteck, Inc., Portland, OR, USA, 42702-102-15) and application of Gonak solution (AKORN, Lake Forest, IL, USA), contact loops were placed on the surface of the cornea. The mice were tested using a series of light flashes of increasing intensity (−20 dB = 0.025 cd·s/m^2^; 0 dB = 2.5 cd·s/m^2^; 5 dB = 7.9 cd·s/m^2^; 10 dB = 25 cd·s/m^2^; 15 dB = 79.1 cd·s/m^2^, 18 dB = 157.73 cd·s/m^2^).

### 4.3. Histology (H&E Staining)

At p45, rd10 eyeballs were enucleated and placed in 4% paraformaldehyde (PFA) overnight at 4 °C. The fixed eyes were washed with PBS and immersed in a series of sucrose solutions in 1X PBS in the following order/time: 10% for 1 h, 20% for 1 h, and 30% overnight. The eyes were embedded in tissue-tek O.C.T. compound (VWR: 25608-930, Visalia, CA, USA) and kept at −80 °C until sectioning. Eyes were then cut to 10 μm sections using a cryostat tissue-sectioning system (Leica CM 1510S; Leica, Buffalo Grove, IL, USA). The sections were stained with hematoxylin and eosin or processed for immunohistochemistry. A blind-to-results investigator calculated mouse photoreceptor nuclei.

### 4.4. Immunohistochemistry

For the preparation of retinal flat mounts, at p30, the eye balls were placed in 4% paraformaldehyde (PFA) for 30 min at 4 °C. After incubation and washing in PBS, the retinas were isolated under a dissecting microscope to perform four incisions from the periphery toward the center. The eyecups were then blocked with 1X PBS containing 0.2% triton-X100 for 2 h at room temperature, followed by incubation with fluorescent PNA lectin fluorescence antibody in IHC buffer (1:1000 diluted, 1% BSA, 0.025% Triton-X100 in PBS) overnight at 4° C. The eyecups were then washed 3 times in 1X PBS and transferred to a glass slide to mount with Fluoromount-G^®^ (Cat#: 0100-01, SouthernBiotech, Birmingham, AL, USA). Images were taken using a NIKON AX-R confocal microscope (Nikon Instruments Inc., Melvine, NY, USA). Retinal sections were blocked in 1X PBS containing 1% bovine serum albumin (BSA, Cat#: 9048-46-8, Fisher Scientific, Hampton, NH, USA), 5% normal donkey serum (Cat#: s30-100 mL, MilloporeSigma, Burlington, MA, USA), and 0.025% triton-X100 for 2 h at room temperature to block nonspecific binding, followed by incubation with primary antibodies including anti-RHO (Cat#: ab155097, Abcam, Boston, MA, USA), anti-PDE6 (Cat#: PA1-722, Invitrogen), and PNA lectin overnight at 4 °C in a humidity chamber. After three washes with 1X PBS, the sections were incubated with secondary antibody (anti-rabbit IgG H&L Alexa Fluor^®^55) for 2 h at room temperature. After washing with 1X PBS, sections were stained with DAPI (Cat#: D3571, Invitrogen) in the dark for 10 min and mounted with Fluoromount-G^®^ (SouthernBiotech Cat#: 0100-01, Birmingham, AL, USA).

### 4.5. Retinal Explants

Retinas from p45 P23H RHO mice were carefully isolated from enucleated eyes and placed in neurobasal serum-free medium (Neurobasal-A, 10888022; Invitrogen, Carlsbad, CA, USA). The medium was supplemented with 2% B27 (0080085-SA; Invitrogen), 1% N_2_ (17502-048; Invitrogen), 2 mM GlutaMAX (35050038; Invitrogen), and 100 units/mL penicillin–100 μg/mL streptomycin (P4333; Sigma-Aldrich Corp., St. Louis, MO, USA). Retinal explants were then maintained at 37 °C in a 5% CO_2_ atmosphere. Afatinib treatment at a concentration of 100 μM was applied for 48 h.

### 4.6. Transfection and Treatment of 661W Cells

The 661W cells were treated with 5 μM afatinib in 15 cm cell culture dish for 24 h. Additionally, we performed an experiment in which we transfected 661W cells with a vector encoding TRIB3 protein fused with GFP (pTRIB3-GFP) as previously described [14]. After 24 h, transfected cells were treated with afatinib for the next 24 h to perform the quantitation of fluorescence.

### 4.7. PDE Activity Assay

Phosphodiesterase activity was measured using the PDE-GloTM Phosphodiesterase Assay KIT (V1361-Promega, Mandison, WI, USA) according to the manufacturer’s instructions. Briefly, mouse retinas were dissected at p30 and homogenized in an immunoprecipitation (IP) lysis buffer containing 25 mM TRIS HCl pH 7.4, 150 mM NaCl, 1 mM EDTA, 1% NP-40, and 5% glycerol without sodium dodecyl sulfate (SDS) detergent (Cat#: 87787, ThermoFisher Scientific, Waltham, CA, USA). Protein concentrations were estimated using the Bio-Rad protein assay (Cat#5000001, Hercules, CA, USA). Then, 50 μg of protein lysate was used for each PDE assay. To initiate the PDE reaction, 20 μM of cGMP solution was added to the wells of 96-well plate and incubated at room temperature for 20 min at room temperature. Kinase-Glo^®^ was then added to the reactions and incubated for 10 min at room temperature. Finally, the luminescence was measured by a plate-reading luminometer (PerkinElmer Victor3 V, Shelton, CT, USA).

### 4.8. Western Blot

The 611W cell lysate and retinal protein extracts from CPP-injected and afatinib-treated rd10 mice were prepared as described [14]. Briefly, 40 μg of each protein lysate was separated by polyacrylamide gel electrophoresis. The detection of EGFR and TRIB3 was performed using primary anti-EGFR (Cat#: ab40815, Abcam, Boston, MA, USA) and anti-TRIB3 (Cat#: sc-390242, Santa Cruz, Dallas, TX, USA) antibodies, followed by secondary HRP antibody (anti-Rabbit Cat#926-80011, anti-Mouse Cat#926-80010 Licor, Lincoln, NE, USA) and imaging the blot using the LI-COR Odyssey Imager (Lincoln, NE, USA).

### 4.9. Statistics

Data were analyzed using either a two-tailed unpaired Student *t*-test, or one-way or two-way ANOVA. Differences were considered significant when *p* < 0.05. All statistical data were calculated using GraphPad Prism 9 software.

## Figures and Tables

**Figure 1 ijms-25-04716-f001:**
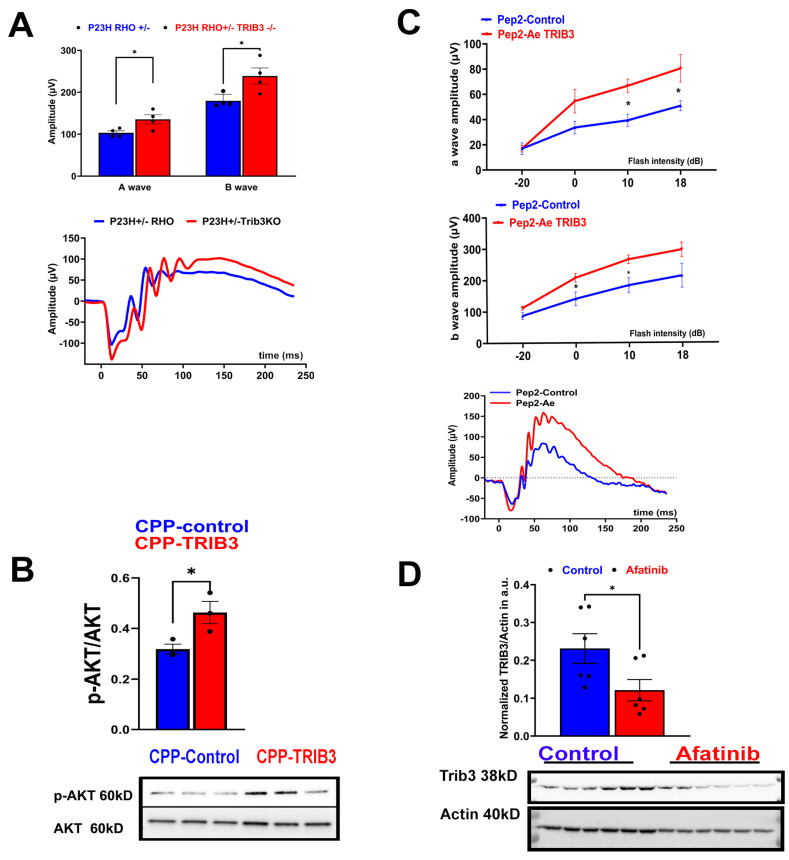
Targeting TRIB3 in P23H RHO retinas. (**A**) TRIB3 ablation in P23H RHO mice leads to improvements of scotopic a- and b-wave ERG amplitudes at p45. The ERG waveforms are shown on the bottom. (**B**) Cell-penetrating peptides targeting TRIB3 activate AKT and protect mice with RD from functional loss. Rd10 mouse retinas were injected with CPPs, Pep2-contr, and Pep2-Ae, delivered by IVT injections. Six hours post-injection, retinas were harvested, and the retinal protein extracts were prepared to run a western blot. A quantitation of normalized p-AKT is shown on the upper panel. Images of the membrane probed with anti-p-AKT and anti-AKT antibodies are shown on the bottom panel (*n* = 3). (**C**) P23H RHO mice were injected with Pep2-cont and Pep2-Ae in the left and right eyes, respectively, on p15 and p25. Scotopic ERG amplitudes were registered at p60 (*n* = 5; * *p* < 0.05). The representative a- and b-wave amplitudes registered with the control and experimental eyes are shown on the bottom. (**D**) Treatment of p45 P23H RHO retinal explants cultured in a medium supplemented with 100 μM afatinib for 48 h showed marked reduction in TRIB3. Images of the membrane probed with anti-TRIB3 and anti-actin antibodies are shown on the bottom (*n* = 6).

**Figure 2 ijms-25-04716-f002:**
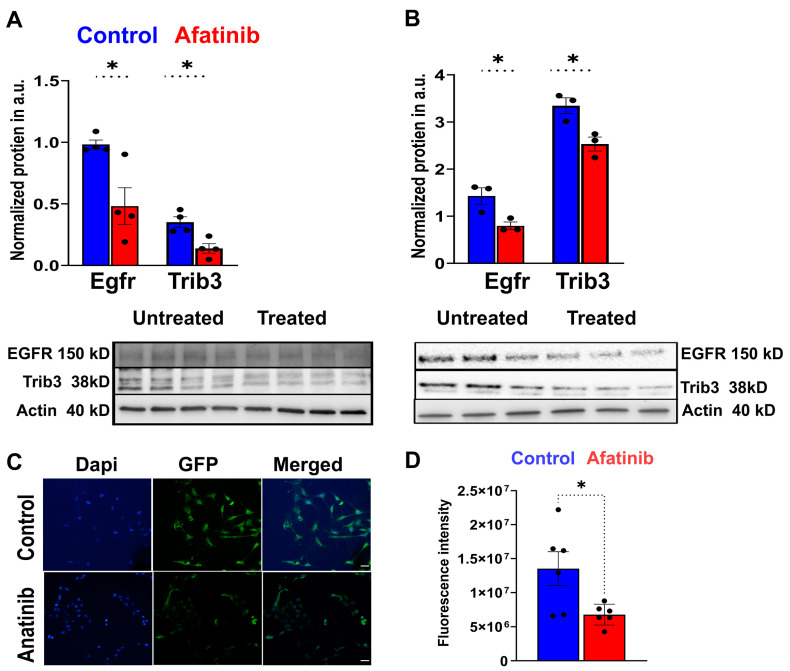
Treatment of 661W cells transfected with fused TRIB3 demonstrated a reduction in the TRIB3 protein level. (**A**) 661W cells were treated with 5 μM for an additional 24 h (*n* = 4). The cells were harvested, and protein extracts from the treated and untreated cells were prepared. Both EGFR and TRIB3 levels were reduced by over 50% in the afatinib-treated cells. (**B**) Rd10 mice were treated daily with afatinib as described in the methods (*n* = 3). Retinas were harvested at p25 to prepare protein extracts and run a western blot. Significant reductions in both EGFR and TRIB3 were observed in the retinas of treated mice (*n* = 5; * *p* < 0.05). (**C**) 661W cells were transfected with pTRIB3-GFP plasmid for 24 h, and then treated with afatinib for an additional 24 h. The fluorescence emitted by transfected and treated cells was calculated using images obtained with confocal microscopy and processed with ImageJ software 1.53k (*n* = 10). Scale bar: 25 μm. (**D**) Intensity per cell was calculated and plotted using GraphPad software 10. Over a 50% reduction in fluorescence emitted by cells expressing TRIB3-GFP and treated with afatinib was observed.

**Figure 3 ijms-25-04716-f003:**
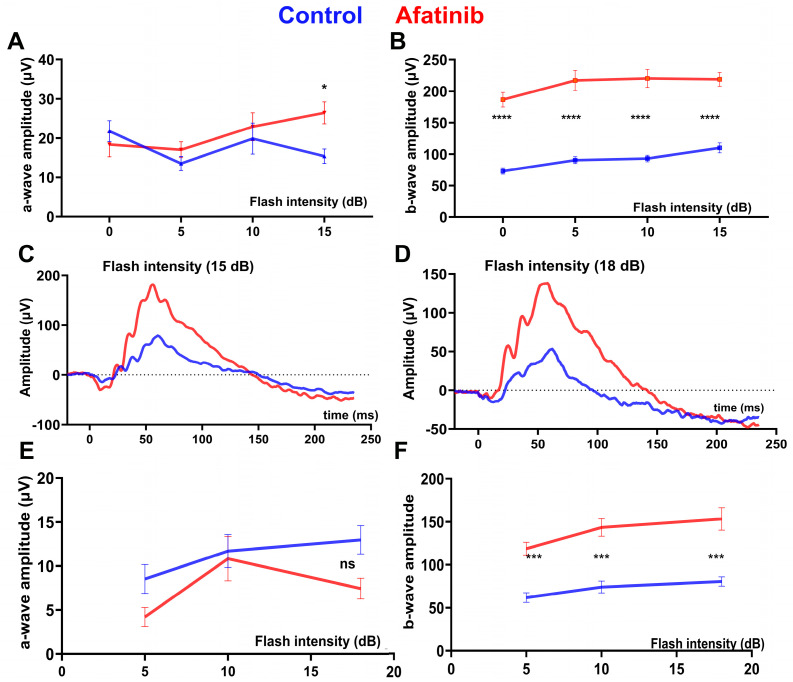
Afatinib improves retinal function in rd10 mice at p30. Animals were treated daily with afatinib according to the specified methods. The averaged scotopic a-wave (**A**) and b-wave (**B**) amplitude responses to a series of flash energy intensities (0, 5, 10, and 15 dB) are presented. (**C**) Representative scotopic ERG traces at an intensity of flash energy set at 15 dB are shown. Subsequently, we recorded photopic ERG responses in the treated rd10 mice. (**D**) Representative photopic ERG traces at an intensity of flash energy set at 18 dB are displayed. The averaged photopic a-wave (**E**) and b-wave (**F**) amplitude responses to a series of flash energy intensities (5, 10, 15, and 18 dB) are shown. Data are presented as mean ± SEM and were analyzed using two-way ANOVA. Statistical significance is indicated as * *p* < 0.05; *** *p* < 0.001; **** *p* < 0.0001; ns = not significant.

**Figure 4 ijms-25-04716-f004:**
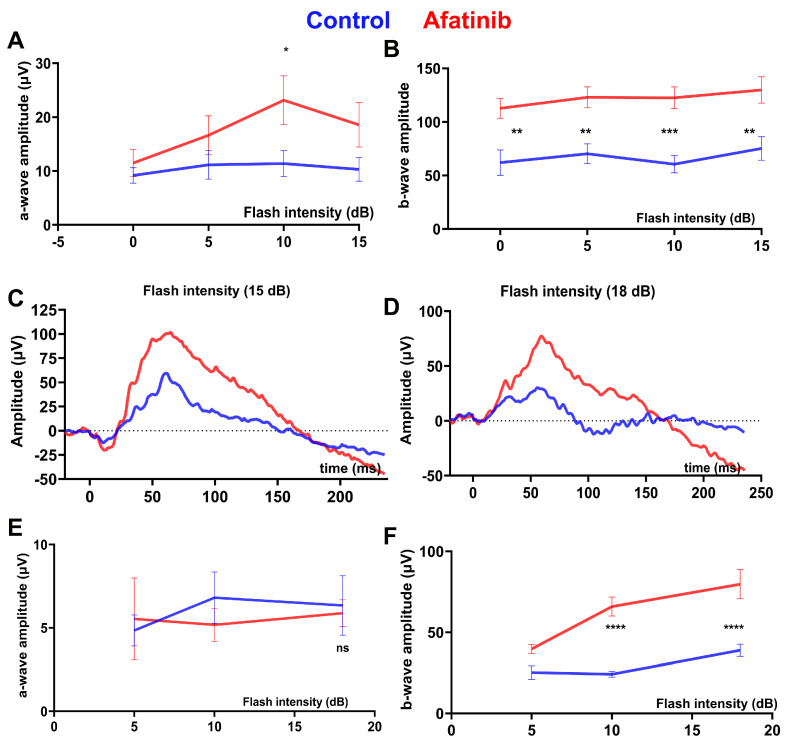
Afatinib improves retinal function in rd10 mice at p45. Animals were treated daily with afatinib according to the specified methods. The averaged scotopic a-wave (**A**) and b-wave (**B**) amplitude responses to a series of flash energy intensities (0, 5, 10, and 15 dB) are presented. (**C**) Representative scotopic ERG traces at an intensity of flash energy set at 15 dB are shown. Subsequently, we recorded photopic ERG responses in the treated rd10 mice. (**D**) Representative photopic ERG traces at an intensity of flash energy set at 18 dB are displayed. The averaged photopic a-wave (**E**) and b-wave (**F**) amplitude responses to a series of flash energy intensities (5, 10, 15, and 18 dB) are shown. Data are presented as mean ± SEM and were analyzed using two-way ANOVA. Statistical significance is indicated as * *p* < 0.05; ** *p* < 0.01, *** *p* < 0.001; **** *p* < 0.0001; ns = not significant.

**Figure 5 ijms-25-04716-f005:**
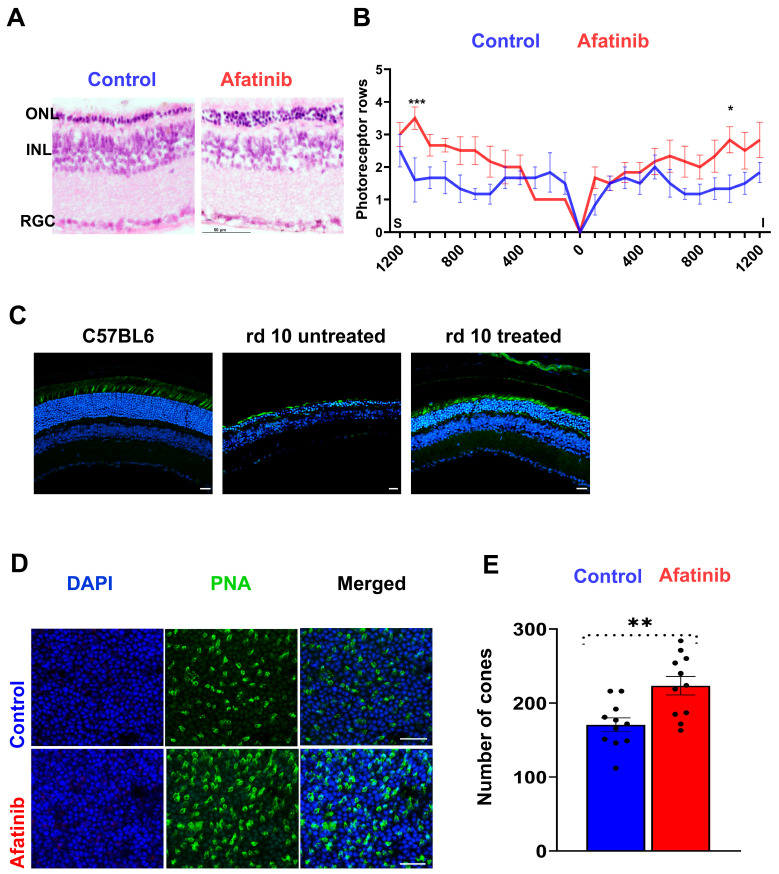
Afatinib demonstrated delayed photoreceptor cell loss observed in treated rd10 retinas. (**A**) Representative images of rd10 retinas treated with PBS (control, left) and afatinib (right), stained with eosin and hematoxylin, are displayed. Scale bar: 50 μm. (**B**) Photoreceptor nuclei were quantified across the retinas and plotted as spider grams using GraphPad. An increase in the number of photoreceptor nuclei was observed. (**C**) Representative images of cryostat-preserved retinal sections from treated mice stained with PNA for cone labeling are presented. (**D**) Flat-mounted retinas were prepared and stained with PNA for cone labeling and DAPI for nuclei detection. Scale bar: 25 μm. (**E**) Cone counting was performed from the same regions (retinal samples from 6 animals, *n* = 11). ONL: Outer Nuclear Layer, INL: Inner Nuclear Layer, RGC: Retinal Ganglion Cell Layer. The data are presented as mean ± SEM; * *p* < 0.05; ** *p* < 0.01; *** *p* < 0.001.

**Figure 6 ijms-25-04716-f006:**
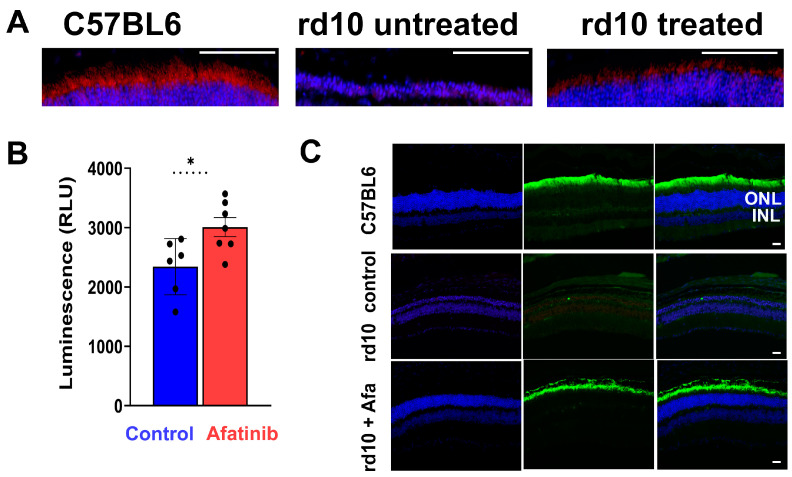
Treatment with afatinib boosts PDE6β in treated rd10 retinas. (**A**) Cryoprotected retinal sections were probed with anti-PDE6β primary antibody to show PDE6β expression in rd10 mice treated/untreated with afatinib versus wild-type C57BL/6J. Representative images are shown. Please also see Appendix A. Scale bar: 25 μm. (**B**) PDE6 activity was measured by PDE-GloTM Phosphodiesterase Assay KIT. (**C**) Detection of rhodopsin with ID4 antibody. A boost in rhodopsin expression was detected in treated rd10 retinas. Representative images are shown. Scale bar: 25 μm. The data are shown in mean ± SD and were analyzed by *t*-test. The significance is depicted as * *p* < 0.05.

## Data Availability

All data resulting from our study are included in this published article and its Appendix A. Primary data are available from the communicating authors upon request.

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
