# Peer review of "Development of TRIB3-Based Therapy as a Gene-Independent Approach to Treat Retinal Degenerative Disorders"

_ijms, 2024, doi:10.3390/ijms25094716_

Round 1
Reviewer 1 Report
Comments and Suggestions for Authors
Dear Editors,
The revised manuscript “Development of TRIB3-Based Therapy as a Gene-Independent Approach to Treat Retinal Degenerative Disorders” presents several errors at multiple levels, despite displaying original data. Therefore, I do not recommend this article for publication.
Critical issues:
Experimental design: Lines 86-89: “Using two models of RD, rd10 and P23H RHO, we employed a pharmacological targeting EGFR-TRIB3 and delivery of cell-penetrating peptides targeting TRIB3→AKT interactomes. The application of both strategies reduced TRIB3 activity and significantly mitigated retinal degeneration in these mice.” None of the presented experiments are repeated on both models. I.e. In figure 1 the author utilizes Pep2Ae as a strategy to inhibit association of TRIB3 with Akt and promote its phosphorylation; in the rd10 animal (Fig. 1A) Pep2Ae was used to show increased pAkt in western-blots from retinal homogenates. In figure 1 B, C and D Pep2Ae is used in P23H animals and retinal function is assessed by electroretinography. In the following results, the drug afatinib is used with the objective to also suppress TRIB3, however, this interaction is not specific, as the authors themselves describe in the paper, afatinib can suppress the activity of TRIB2. Besides this, afatinib has several other targets other than TRIB3: “Like other protein kinase inhibitors, the mechanism of action of afatinib is to irreversibly inhibit the epidermal growth factor receptor (EGFR), human epidermal growth factor receptor 2 (HER2), and HER4.[1] EGFR and HER2 are part of the receptor tyrosine kinase family” (Afatinib - StatPearls - NCBI Bookshelf (nih.gov)). Therefore, the effects of using Pep2Ae and Afatinib are not directly comparable, and additional experiments would be required if the author is to affirm that modulation of TRIB3 has similar effects on P23H and rd10 animals. Either in the form of additional experiments using Pep2Ae on rd10 mice (such as electroretinographies, as used on P23H), or experiments with afatinib on P23H animals.
Use of non-representative images: Fig 5A - The image is not representative of the plotted graphs. The untreated retina shows 1 layer of photoreceptors while the treated one has 5-7 rows of photoreceptors. In the graph the best results observed were preservation of 4 rows of photoreceptors at most. Fig 5D - This image is not representative of the quantification graph. The afatinib treated group clearly has less PNA positive cells than the control group.
References: Several references are incorrectly used, especially on the introduction section. Either by citing articles that do not demonstrate what is being affirmed, or by clearly citing incorrect references. In example:
Lines 57-59: “Moreover, strategies such as constitutive activation of mTORC1 in cones via 57 disruption of PTEN or TSC significantly improved retinal cell metabolism and survival. [14, 15]”
Reference 14 supresses mTOR by ablation of RAPTOR and RICTOR, and not inducing its expression via supression of PTEN. Besides, it observes no effect on cone cell survival, only its metabolism.
Lines 59-62: “Additionally, targeting the downstream mTORC1 effector S6K1 by subretinal Adeno-Associated virus (AAV)-mediated S6K1 delivery significantly improved rod photoreceptor survival in rd10 mice, a model of retinal degeneration expressing mutant PDE6β”. [6]
Which cites: “Translational attenuation and retinal degeneration in mice with an active integrated stress response”
This happens across several references used in the introduction section.
Among these problems, other minor issues are also present. Therefore, I suggest this paper to be rejected from publication in its current state.
Comments on the Quality of English LanguageEnglish is not a problem in this paper.
Author Response
Dear Reviewer,
Thank you very much for dedicating your time to reviewing our manuscript. We greatly appreciate your valuable comments and suggestions. We have diligently reviewed all of your feedback and have integrated suggested changes into our resubmitted manuscript. We hope our editing and changes improved the quality of the manuscript. We look forward to hearing your thoughts on the revised manuscript and hope that it will meet with your approval.
Q#1: Critical Issues: Lines 86-89: “Using two models of RD, rd10 and P23H RHO, we employed a pharmacological targeting EGFR-TRIB3 and delivery of cell-penetrating peptides targeting TRIB3→AKT interactomes. The application of both strategies reduced TRIB3 activity and significantly mitigated retinal degeneration in these mice.” None of the presented experiments are repeated on both models. I.e. In figure 1 the author utilizes Pep2Ae as a strategy to inhibit association of TRIB3 with Akt and promote its phosphorylation; in the rd10 animal (Fig. 1A) Pep2Ae was used to show increased pAkt in western-blots from retinal homogenates. In figure 1 B, C and D Pep2Ae is used in P23H animals and retinal function is assessed by electroretinography. In the following results, the drug afatinib is used with the objective to also suppress TRIB3, however, this interaction is not specific, as the authors themselves describe in the paper, afatinib can suppress the activity of TRIB2. Besides this, afatinib has several other targets other than TRIB3: “Like other protein kinase inhibitors, the mechanism of action of afatinib is to irreversibly inhibit the epidermal growth factor receptor (EGFR), human epidermal growth factor receptor 2 (HER2), and HER4.[1] EGFR and HER2 are part of the receptor tyrosine kinase family” (Afatinib - StatPearls - NCBI Bookshelf (nih.gov)). Therefore, the effects of using Pep2Ae and Afatinib are not directly comparable, and additional experiments would be required if the author is to affirm that modulation of TRIB3 has similar effects on P23H and rd10 animals. Either in the form of additional experiments using Pep2Ae on rd10 mice (such as electroretinographies, as used on P23H), or experiments with afatinib on P23H animals.
Answer: We appreciate the reviewer's suggestion regarding the specific targeting of TRIB3 by afatinib. It's worth noting that our experiment with TRIB3 GFP transfected cells treated with afatinib indeed shows a reduction in TRIB3 levels. While we cannot definitively conclude whether afatinib directly binds to TRIB3 or reduces it through degradation of EGFR-TRIB3 complex, it's noteworthy that TRIB3 binds to EGFR and afatinib inhibits EGFR. Therefore, we propose that the reduction in TRIB3 might be due to the degradation of the EGFR-TRIB3 complex mediated by afatinib. We discussed this point in the revised manuscript. We also thank the reviewer for the comment regarding the need for additional experiments to bolster the hypothesis on therapeutic targeting of TRIB3 in degenerating retinas. In response, we have included data on P23H RHO TRIB3 knockout mice, which demonstrate improved scotopic a- and b-wave ERG amplitudes (Fig. 1). Additionally, we have incorporated ex vivo experiments with cultured P23H RHO retinas treated with afatinib, revealing a significant reduction in TRIB3 levels (Fig.1). These findings, coupled with results obtained from TRIB3 knockout mice, suggest that afatinib treatment could potentially benefit both P23H RHO and rd10 mice. Regarding the treatment of P23H RHO mice with afatinib, we have to admit that, at present, we are unable to conduct long-term experiments with these mice due to the additional time and resources required for such longitudinal studies.
Q#2: Use of non-representative images: Fig 5A - The image is not representative of the plotted graphs. The untreated retina shows 1 layer of photoreceptors while the treated one has 5-7 rows of photoreceptors. In the graph the best results observed were preservation of 4 rows of photoreceptors at most. Fig 5D - This image is not representative of the quantification graph. The afatinib treated group clearly has less PNA positive cells than the control group.
Answer: Thank you for careful reading. The figure 5 has been edited, and representative images have been replaced with improved ones.
Q#3: References: Several references are incorrectly used, especially on the introduction section. Either by citing articles that do not demonstrate what is being affirmed, or by clearly citing incorrect references.
Answer: Thank you for your attention to detail. All references have been verified and arranged in the correct order.
Reviewer 2 Report
Comments and Suggestions for Authors
This study report the results of a novel gene independent treatment based on the use of TRIB 3. The animal model tested was the rd10 mouse. The authors show significant protective effect of this treatment.
It would be important to understand if the protective effect is exerted on phototransduction. Therefore, it may be helpful to record ERG a-waves and look at the response kinetics according to Lamb's and Pugh's model.
Author Response
Dear Reviewer,
Thank you very much for dedicating your time to reviewing our manuscript. Your valuable comments and suggestions are greatly appreciated.
Q#1: It would be important to understand if the protective effect is exerted on phototransduction. Therefore, it may be helpful to record ERG a-waves and look at the response kinetics according to Lamb's and Pugh's model.
Answer: While we agree that exploring the phototransduction and the proposed protocol would be intriguing, unfortunately, we are not familiar with this method at this time.
Round 2
Reviewer 2 Report
Comments and Suggestions for Authors
No further comments